# Recent Advances on Electrochemical Sensors for Detection of Contaminants of Emerging Concern (CECs)

**DOI:** 10.3390/molecules28237916

**Published:** 2023-12-03

**Authors:** Chinchu Gibi, Cheng-Hua Liu, Sambandam Anandan, Jerry J. Wu

**Affiliations:** 1Department of Environmental Engineering and Science, Feng Chia University, Taichung 407, Taiwan; chinchugibi1@gmail.com (C.G.); chhengliu@fcu.edu.tw (C.-H.L.); 2Department of Chemistry, National Institute of Technology, Trichy 620015, India; sanand99@yahoo.com

**Keywords:** Contaminants of Emerging Concern, electrochemical sensors

## Abstract

Contaminants of Emerging Concern (CECs), a new category of contaminants currently in the limelight, are a major issue of global concern. The pervasive nature of CECs and their harmful effects, such as cancer, reproductive disorders, neurotoxicity, etc., make the situation alarming. The perilous nature of CECs lies in the fact that even very small concentrations of CECs can cause great impacts on living beings. They also have a nature of bioaccumulation. Thus, there is a great need to have efficient sensors for the detection of CECs to ensure a safe living environment. Electrochemical sensors are an efficient platform for CEC detection as they are highly selective, sensitive, stable, reproducible, and prompt, and can detect very low concentrations of the analyte. Major classes of CECs are pharmaceuticals, illicit drugs, personal care products, endocrine disruptors, newly registered pesticides, and disinfection by-products. This review focusses on CECs, including their sources and pathways, health effects caused by them, and electrochemical sensors as reported in the literature under each category for the detection of major CECs.

## 1. Introduction

A booming population and the ever-increasing demands of products and facilities have induced the massive growth of industries and production. With industrialization and technological advancements, pollution of natural resources has become much evident in air, water, and soil. Previously unheard-of diseases are being reported alongside new chemicals being added to the list of harmful contaminants.

Contaminants of Emerging Concern (CECs), Emerging Contaminants (ECs), Persistent Organic Pollutants (POPs), and Persistent, Bioaccumulative, and Toxic (PBT) are some of the novel terms that have recently attracted much attention. A Contaminant of Emerging Concern (CEC) was defined by Diamond et al. as a chemical for which there are increasing concerns regarding its potential risks to humans and ecological systems, including endocrine disruption and neurotoxicity [1]. Sauve et al. defined CECs as naturally occurring, manufactured, or manmade chemicals or materials which have now been discovered or are suspected to be present in various environmental compartments and whose toxicity or persistence is likely to alter the metabolism of a living being significantly [2]. To solve the ambiguity between Emerging Contaminants (EC) and CECs, ECs are contaminants which have appeared recently. It is more of a relative term, as those contaminants regarded as ECs a decade ago may no longer be one in the present. However, CECs are contaminants which have been in the environment for a while, but for whom the concerns have been raised recently. Also, a well-known, already regulated contaminant may reattain the emerging position when new information about the contaminant emerges and regulations need to be revised [3]. At times, ECs and CECs are used interchangeably. Pharmaceuticals, Personal Care Products (PCPs), Endocrine Disruptors (EDs), illicit drugs, newly registered pesticides, Disinfection By-Products (DBPs), and Per- and Polyfluoroalkyl substances (PFAs) are commonly reported CECs.

Although CECs have been polluting the ecosystem for a long time, the amounts and types of CECs have shown a tremendous increase in the past 50 years [4]. CECs are identified at very low concentrations of ng/L to µg/L in surface water, waste water, and drinking water [5]. However, the nature of bioaccumulation of CECs makes the scenario alarming. Bioaccumulation is the build-up of CECs in an organism. Some mixtures of CECs bioaccumulated are more toxic than single CECs [6]. Unlike conventional pollutants, CECs are rarely globally regulated, and other risks related to CECs include endocrine disruption [7], carcinogenicity [8], neurotoxicity, and genotoxicity [9]. Traditional water treatment methods, like activated carbon treatment and reverse osmosis, failed to remove CECs from water [10]. Thus, it is very important to develop an efficient technique to sense CECs and provide safe food and drinking water. Electrochemical sensors, which can detect analytes at a very low concentrations, i.e., the picomolar range, are a better option for CEC detection as they are present in very low concentrations in the environment. Also, electrochemical sensors are a cost-effective option, while still being able to rapidly detect CECs [11]. Electrochemical sensors provide a sensitive, selective, and effective platform to detect CECs in water, food, and serum samples. This review aims to explain about CECs, their sources, the toxic effects of different CEC classes, and electrochemical sensors developed for the detection of these CECs.

## 2. Sources and Toxicity of Contaminants of Emerging Concern (CECs)

Industrial wastewater, agriculture/livestock runoff, and household and hospital effluents are the major sources of CECs, whereas water and soil are the sinks of CECs [12]. Pharmaceuticals, personal care products, and pesticides are common CECs found in industrial effluents. Pesticides used to improve crop productivity are the most common CECs found in agriculture runoff. Livestock runoffs include pharmaceuticals and steroid hormones used in livestock. Hospital and domestic effluents might contain pharmaceuticals, endocrine disruptors, and disinfection by-products [13]. Depending on the physicochemical properties of CECs, they seep into the soil, pollute ground water, or adsorb into the soil, thus contaminating the soil and ground water. Some other CECs get discharged into surface water, leading to surface water contamination [14]. Plants are the major transporters of CECs as they are the primary producers in the food chain [15]. From plants, they get transferred to higher trophic levels and may lead to bioaccumulation.

Unfortunately, CECs have become an integral part of our daily lives. Unregulated use of pharmaceuticals and personal care products has resulted in the release of a large number of CECs, and the unchecked use of insecticides and pesticides to improve crop yields has only added to the situation. The rebellious use of illicit drugs has also increased. These can cause reproductive damage, endocrine disorders, allergies, neurotoxicity, bioaccumulation effects, growth inhibition effects, hormone interference, disrupted physiological processes, increased cancer risks, affected metabolism, transport and synthesis of endogenous hormones and specific receptors [16], and biochemical toxicity to invertebrates, amphibians, and fish [17]. The major classes of CECs, representative compounds, sources, and effects are shown in Table 1.

With the increasing awareness of the toxicity of CECs hidden in our day-to-day products, their detection is also growing in importance. The EPA has developed criteria to assess and manage the risks of CECs in aquatic environments due to their extensive use, chemical persistence, consequences in nature, and the concerns of the public [18]. The Safe Drinking Water Act (SDWA), Toxic Substance Control Act (TSCA), Comprehensive Environmental Response, Compensation and Liability Act (CERCIA), and Clean Water Act (CWA) raised the interest of authorities to address the issue of CECs [19]. CECs may be present in very low concentrations, but can cause serious effects on living beings and nature [20]. For example, fluoxetine at a concentration of 546 ng/L can be fatal to aquatic species [21] and ibuprofen at a minimum concentration of 250 ng/L can cause endocrine disruptions to aquatic species [22]. Therefore, a quick, selective, and sensitive method is necessary for the detection of CECs.

**Table 1 molecules-28-07916-t001:** Classes of CECs, their sources, and toxic effects.

Classes of CECs	Description/Categories	Representative Compounds	Sources	Effects
Pharmaceuticals	Anti-inflammatories, analgesics, antibiotics, antidepressants, lipid lowering agents, antihistamines, β-blockers	Diclofenac, Norfloxacin, Acetylsalicylic acid, Sodium salicylate, Oxaprozin, Ibuprofen, Indomethacin, Acetaminophen, Carbamazepine, Promethazine hydrochloride, Norfloxacin	Over the counter (OTC) Pharmaceuticals, Prescribed Pharmaceuticals	Cancer, liver and kidney failure [23], neurotoxicity, cardiovascular risks, drug-induced hepatoxicity [24]
Illicit Drugs	Non-prescribed drugs	Cocaine, Morphine, Codeine, Amphetamine, MDMA, 6-acetylmorphine	Psychotropic drugs	Neurotoxic effects [25], hypertension [26], low blood pressure, respiratory problems [27], coma [28]
Personal Care Products (PCPs)	Cosmetics, daily care products and fragrances, plasticizers, synthetic musks, UV filters, preservatives	Methyl paraben, Ethyl paraben, Triclosan, Vanillin, Triclosan	Toothpastes, lotions, fragrances, cosmetics, soaps, shampoos	Endocrine effects, reproductive malfunctions [29], contact dermatitis, breast cancer [30], affects CNS [31]
Endocrine-Disrupting Chemicals (EDCs)	Bisphenols (BPs), polychlorinated biphenyls (PCBs), phthalate esters, alkylphenols, natural and synthetic estrogens	BPA, BPS, Diethylstilbestrol, Estradiol, Phthalates	Chemicals used as solvents or lubricants, their byproducts, plasticizers, electronic materials	Reproductive issues, neurological damages, cardiovascular diseases, diabetes [32], fertility defects, sexual abnormalities, cancer [33]
Newly Registered Pesticides	New pesticide or chemicals or different uses of existing chemicals	Imidacloprid (IDP), Thiamethoxam (TMX)	Pesticides used to improve the crop yield	Cancer, effects on immune system, reproductive system, respiratory system [34]
Disinfection Byproducts (DBPs)	Chlorates, chlorites, bromates, trihalomethanes (THMs), and haloacetic acids (HAAs)	Chlorites, bromates, Trichloroacetic acid (TCAA), Trichloroacetamide (TCAM)	Disinfectants react with organic matter or manmade contaminants during water disinfection	Genotoxicity, carcinogenicity [35], acquired methemoglobinemia [36], cancer, reproductive defects [37]
Per- and Polyfluoroalkyl substances (PFAs)	Organofluorine chemical compounds that can resist water, oil, and heat	Perfluorooctanoic acid (PFOA), perfluorooctane sulfonate (PFOS), GenX, polyfluoroalkyl phosphates (PAPs), fluorotelomer sulfonic acids (FTSA)	Packaging products, carpets, firefighting foams, paints, semiconductors, carpets	Immunological, cardiovascular, reproductive, developmental, and liver effects [38]

## 3. Electrochemical Sensors for Detection of CECS

Electrochemical sensors are an efficient tool for sensing CECs. Conventional methods, such as Liquid Chromatography [39], Inductively Coupled Plasma Mass Spectroscopy (ICP-MS) [40], Atomic Absorption Spectroscopy (AAS), Atomic Emission Spectroscopy (AES) [41], Atomic Fluorescence Spectroscopy [42], and Inductively Coupled Plasma-Optical Emission Spectrometry (ICP-OES) [43], are quite expensive, time consuming, and require highly skilled professionals. Compared to conventional techniques, electrochemical methods are more sensitive, stable, portable, and economical, allow the simultaneous detection of multiple analytes, and are environmentally benign. Liquid Chromatography–Mass Spectrometry (LC–MS) is a conventional technique. Electrochemical sensors and optical sensors are commonly used sensors; Table 2 compares various aspects of electrochemical sensors, LC–MS, and optical sensors. Voltammetry is the most commonly used electrochemical technique to detect CECs. Voltammetric techniques measure current with the corresponding variation in potential [44]. Linear Sweep Voltammetry (LSV), Cyclic Voltammetry (CV), Differential Pulse Voltammetry (DPV), Anodic Stripping Voltammetry (ASV), Differential Pulse Anodic Stripping Voltammetry (DPASV), Square Wave Anodic Stripping Voltammetry (SWASV), and Linear Sweep Anodic Stripping Voltammetry (LSASV) are the subclasses of voltammetric techniques based on the mode of voltage variation [45].

Electrochemical sensors comprise a three-electrode system consisting of a reference electrode, counter electrode, and working electrode. The working electrode is where the redox reaction of interest takes place. Commonly used working electrodes are Glassy Carbon Electrodes (GCEs) and Screen-Printed Electrodes (SPEs). SPEs which perform as working electrodes alone, and one that combines all the three electrodes into a single substrate, are available. Features such as portability, low cost, disposability, and omission of electrode polishing have made SPEs popular. SPEs also prevent any possible fouling from analytes used previously as they are disposable [46]. Modification of the working electrode can improve the performance of the sensor. The modification can involve enzymes, DNA, or electrocatalysts. In electrocatalyst-based sensors, the electrocatalyst coated on the surface of working electrode undergoes a redox reaction with the analyte of interest (CECs). The redox reaction between the CEC and electrocatalyst results in a corresponding electrical signal, from which quantitative and qualitative information about the CEC can be deciphered. Also, the electrocatalyst may be selective for a particular CEC, which highlights another important characteristic of electrochemical sensors, which is selectivity [47]. The properties of the electrocatalyst, such as chemical structure and morphology, play an important role in the sensitivity of electrochemical sensor [48]. Therefore, the use of a suitable electrocatalyst is vital for the development of a good electrochemical sensor.

### 3.1. Electrochemical Sensors for Detection of Pharmaceuticals

Pharmaceuticals are one of the most widely occurring CECs. Pharmaceuticals are substances that are used for therapeutic, preventive, and diagnostic purposes, and do not include recreational drugs like cocaine [49]. Anti-inflammatories, analgesics, antibiotics, antiepileptics, antidepressants, lipid lowering agents, antihistamines, and β-blockers belong to the class of pharmaceuticals [23,50]. The presence and distribution of pharmaceuticals have become an issue of major concern [24,51]. The global usage of medicine achieved 4.5 trillion doses by 2020, when half of the world population consumed >1 dose/person/day of drugs [52].

Pharmaceutical products have contaminated water sources, soil, and sediments [53]. They have the nature of persistent contaminants, which are resistant to complete degradation [54]. This leads to pollution of water resources and soil, thus harming aquatic life and other organisms. The presence of pharmaceuticals in soil can affect the biological balance in the soil [55] and thus threaten food safety [56]. Pharmaceutical products in the aquatic environment have caused health effects such as cancer, liver, and kidney failure [57]. “Drug-induced hepatoxicity” implies liver damage driven by drugs, which can directly or indirectly damage liver cells, which is a vital organ for carbohydrate, protein, and fat metabolism, detoxification, drug transformation, bile secretion, and vitamin storage [58]. Therefore, a number of electrochemical sensors have been developed for the detection of pharmaceutical products.

Acetaminophen (N-(4-hydroxyphenyl)acetamide), commonly known as paracetamol (PA), is a painkiller used to reduce fever and pain worldwide. Excess use of PA can affect the liver and kidneys. Spinel vanadium nano ferrite (VFe_2_O_4_)-modified GCEs were developed for the electrochemical sensing of PA in pharmaceutical and biological samples with a limit of detection of 8.20 nM [59]. Simultaneous femtomolar detection of PA, Diclofenac (2-[2-(2,6-dichloroanilino)phenyl]acetic acid, DIC), and Orphenadrine ((RS)-N,N-Dimethyl-2-[(2-methylphenyl)-phenyl-methoxy]-ethanamine, ORP) was achieved using COOH-CNTs/ZnO/NH_2_-CNTs modified on GCEs, where ZnO nanoparticles were trapped in between the COOH-CNT and NH_2_-CNT [60]. DIC, an antihistamine, may affect the central nervous system, and ORP, an antirheumatic drug, can cause heart attacks. The limits of detection of PA, DIC, and ORP by COOH-CNTs/ZnO/NH_2_-CNTs/GCE were 46.8, 78, and 60 fM, respectively. Detection of DIC in human urine samples was reported using exfoliated graphene-supported cobalt ferrite (EGr-Co_1.2_Fe_1.8_O_4_)-modified SPCEs, synthesized by an ultrasonication method as shown in Figure 1A [61]. The EGr-Co_1.2_Fe_1.8_O_4_/SPCE outperformed the Co_1.2_Fe_1.8_O_4_/SPCE, Gr-Co_1.2_Fe_1.8_O_4_, and bare SPCE in CV analysis for the detection of DIC as shown in Figure 1B. With the increase in concentration of DIC, the peak current value also increases (Figure 1C) in DPV analysis, and shows a linear relationship between the concentration of analyte and current value from which a detection limit of 1 nM and sensitivity of 1.059 μA μM^−1^ cm^−2^ were deciphered. Copper–aluminum-layered double hydroxide, homogeneously dispersed over oxidized graphitic carbon nitride-modified GCEs (oxidized g-C_3_N_4_/Cu–Al LDH/GCE), were fabricated for the detection of Diclofenac Sodium (DS). Due to the combined effect of oxidized g-C_3_N_4_ and Cu–Al LDH, a good electrochemical response for DS oxidation and enhanced current was obtained. The sensor could achieve a detection limit of 0.38 μM and linear range of 0.5–60 μM using the DPV technique [62].

Ibuprofen ((RS)-2-(4-(2-methylpropyl)phenyl)propanoic acid, IBU) is the third highly consumed non-steroidal anti-inflammatory drug used to relieve pain, arthritis, and inflammatory diseases. Overdose of IBU can increase the risk of heart attacks, and due to its antiplatelet outcome, it is known as a blood-thinning drug. For the detection of IBU, copper tellurate (Cu_3_TeO_6_) with a 3D stone-like morphology was developed using a wet-chemical method [63]. The Cu_3_TeO_6_/GCE reported a detection limit of 0.017 µM and linear range of 0.02–5 μM and 9–246 μM. Nitrofurazone ((2E)-2-[(5-Nitro-2-furyl)methylene]hydrazine carboxamide, NZ) is an antibiotic used to treat ulcers, bladder cancer, skin inflammation, and gastrointestinal infections. NZ has genotoxic, mutagenic, and carcinogenic effects. Sulfur-doped graphitic carbon nitride with copper tungstate hollow spheres (Sg–C_3_N_4_/CuWO_4_) was synthesized using an ultrasonic method [64]. The Sg–C_3_N_4_/CuWO_4_/GCE exhibited a detection limit of 3 nM and sensitivity of 1.24 µAµM^−1^cm^−2^ towards the detection of NZ in human urine and serum samples. Norfloxacin (1-ethyl-6-fluoro-4-oxo-7-piperazin-1-yl-1H-quinoline-3-carboxylic acid (NFX)) is a fluoroquinolone used to treat human and veterinary infections. NFX can disrupt the endocrine systems of aquatic organisms. A CaCuSi_4_O_10_/GCE was developed for the detection of NFX. The inter-layer spaces of the layered silicate structure offered suitable adsorption sites. In order to improve conductivity, MnO_2_ was also added. The sensor showed a detection limit of 0.0046 μM and linear ranges of 0.01–0.55 and 0.55–82.1 µM [65].

Naproxen sodium (NAP) and Sumatriptan (SUM) are drugs which have shown anti-viral effects against COVID-19. NAP ((+)-(S)-2-(6-methoxynaphthalene-2-yl) propanoic acid) is used to relieve pain and rheumatic disorders. Overdose of NAP can lead to kidney or liver disease. SUM (1-[3-(2-dimethyl aminoethyl)-1 H-indol-5-yl]-N-methylmethanesulfonamide) is a tryptamine-based medicine used to treat migraine headaches. SUM overdose can narrow blood vessels, leading to heart issues like heart attacks. Auxiliary use of NAP and SUM is associated with serotonin syndrome. Multiwalled carbon nanotubes decorated with ZnO, NiO, and Fe_3_O_4_ nanoparticles on Glassy Carbon Electrodes (GCEs) (ZnO/NiO/Fe_3_O_4_/MWCNTs) were developed for the detection of NAP and SUM with a detection limit of 3 nM and 2 nM, respectively [66]. The high electrical conductivity and electroactive surface area of the composite improved the electrooxidation of NAP and SUM. The linear ranges of detection were 4.00 nM to 350.00 μM for NAP and from 6.00 nM to 380.00 μM for SUM. Yet another electrode developed for the simultaneous detection of NAP and SUM was a carbon paste electrode modified with peony-like CuO-Tb^3+^ nanostructures (P-L CuO: Tb^3+^ NS/CPE) [67]. P-L CuO: Tb^3+^ NS/CPE gave linear ranges of 0.01–800 μM and 0.01–700 μM, and detection limits of 3.3 nM and 2.7 nM for the detection of SUM and NAP, respectively.

Vortioxetine (VOR) (1-[2-(2,4-dimethyl-phenylsulfanyl)-phenyl]-piperazine-hydrobromide) is an antidepressant used in the treatment of major depressive disorder. VOR is associated with sexual dysfunction, liver injury, anxiolytic like behavior in children, histopathological damage, nausea, and vomiting [25]. Vortioxetine (VOR) detection was made possible using an electrochemical sensor based on gold nanoparticles/graphene (AuNP_s_@GRP) modification on a GCE [26]. AuNP_s_@GRP synthesized by a one-pot method enhanced the sensitivity of the sensor for the selective detection of VOR. The electrochemical sensor could detect VOR at a detection limit of 50 nM and linear ranges of 0.1–1.0 and 1.0–6.0 μM in pharmaceutical samples. Promathazine hydrochloride (PMHC) ((N, N-dimethyl-1-phenothiazin-10-yl-propan-2-amine hydrochloride) is another drug commonly prescribed for mental illness. PMHC overdose is related to cardiac problems and reproductive dysfunction, which can be fatal. A hybrid of barium tungstate with functionalized carbon black modified on an SPCE (BaWO_4_/*f*-CB/SPCE) was fabricated for the detection of PMHC [68]. BaWO_4_/*f*-CB was synthesized by co-precipitation followed by an ultrasonication method and then drop casted on an SPCE for PMHC sensing as shown in Figure 1D. As compared to a bare SPCE, BaWO_4_/SPCE, and f-CB/SPCE, the BaWO_4_/*f*-CB/SPC gave better oxidation and reduction peak currents (Figure 1E) and the efficiency of the electrode for PMHC sensing was proved. A detection limit of 29 nM was inferred from the DPV analysis with an increase in the concentration of PMHC (Figure 1F) and the resulting linear graph of current and concentration. Carbamazepine (CBZ) (5H-dibenzo [b,f]azepine-5-carboxamide) is an anticonvulsant medicine used for the treatment of bipolar disease, mental disorder, and post-traumatic stress disorder. Growing awareness about the toxicity of CBZ is due to the harmful consequences such as neurotoxicity. A gadolinium vanadate nanostructure decorated on functionalized carbon nanofibers on a Glassy Carbon Electrode (GdVO_4_/*f*-CNF/GCE) was developed for the detection of CBZ in pharmaceuticals and human urine samples [27]. The synergistic effect between GdVO_4_ and *f*-CNF enhanced the electrochemical performance of the sensor. A detection limit of 0.0018 μM and linear range of 0.01–157 μM were achieved using the electrochemical sensor for CBZ detection.

### 3.2. Electrochemical Sensors for Detection of Illicit Drugs

Illicit drugs are non-prescribed drugs or psychotropic substances [69]. International Drug Control Conventions prohibit the production, sale, and use of illicit drugs [70]. Cocaine, benzoylecgonine, morphine, codeine, 6-acetylmorphine, methadone, amphetamine, methamphetamine, and 3,4-methylenedioxyamphetamine are common illicit drugs. According to the World Drug Report 2023 launched by the UN Office on Drugs and Crime (UNODC), over 296 million people used drugs in 2021, which is 23 percent greater than the previous decade. The number of people suffering from drug disorders shot up to 39.5 million, a 45 per cent increase over 10 years [71]. These drugs are consumed by means of ingestion, inhalation, absorption, injection, smoking, dissolution under the tongue, or skin patching. Due to the stimulus that these drugs provide, people use them even though they are dangerous and become addicted to these drugs. These drugs have both mental and physical effects. Driving Under Influence of Drugs (DUID) has led to many road accidents [72]. Drug-assisted sexual harassment is another issue of major concern. The misuse of illicit drugs can have severe neurotoxic effects [73].

Cocaine (Methyl (1R,2R,3S,5S)-3-benzoyloxy-8-methyl-8-azabicyclo [3.2.1] octane-2-carboxylate) is a Central Nervous System (CNS) stimulant. Risks associated with cocaine include hypertension, altered mental status, seizure, chest pain, headache, neurological defects, corneal ulceration and vision loss, HIV, and hepatitis [74]. An octahedral palladium-doped cobaltite composite-modified GCE (Oh-Pd^2+^: Co_3_O_4_-C/GCE) was fabricated for cocaine detection in biological specimens [75]. The Oh-Pd^2+^: Co_3_O_4_-C composite was synthesized using a hydrothermal method. The large surface area and higher electrocatalytic activity of the composite gave a detection limit of 1.3 nM and linear range of 0.01 μM–900.0 μM.

Morphine is an opioid used to relieve moderate to severe pain in cancer patients. Like other opioids, morphine also causes addiction, low blood pressure, and respiratory problems [28]. A polydopamine-functionalized multiwalled carbon nanotube-modified GCE (PDA-f-MWCNT/GCE) was used for the detection of morphine [76]. The PDA-f-MWCNT catalyst raised the electrochemical activity for morphine oxidation. The detection limit obtained using this sensor was 0.06 µM, and the linear range of detection of morphine in human plasma and urine samples was 0.075–75.0 μM.

Amphetamine-type stimulants (ATSs) are a group of synthetic drugs, chemically derived from β-phenethylamine, that stimulates the central nervous system. Amphetamine (A), methamphetamine (MA), 3,4-methylenedioxyamphetamine (MDA), 3,4-methylenedioxymethamphetamine (MDMA), and 3,4-methylenedioxyamphetamine (MDEA, MDE or “Eve”) are common drugs in this category [77]. MDMA is a psychoactive recreational hallucinogenic drug used worldwide. Long-term effects of MDMA include kidney failure, hepatoxicity, and decreased immunity [29]. An electrochemical sensor was developed for the simultaneous detection of morphine and MDMA using carbon nanohorns–chitosan decorated with Pt nanoparticles (CNH–CHI@PtNPs) [78]. A schematic representation of the electrode fabrication for sensing morphine and MDMA is given in Figure 2A. The CNH–CHI@PtNPs/GCE showed a better catalytic performance than the CNH–CHI/GCE and bare GCE (Figure 2B). A linear range of 0.05–25.4 µM and detection limit of 0.02 µmol/L and 0.018 µmol/L were obtained using DPV analysis (Figure 2C) for detection of morphine and MDMA, respectively. MDEA has similar effects as that of MDMA. Fast, on-site detection of MDEA was made possible with an electrochemical sensor using a Carbon-Screen Printed Electrode (C-SPE) which yielded a detection limit of 0.03 μmol L^−1^ and linear range of 2.5 to 30.0 μmol L^−1^ [30].

Oxycodone (14-hydroxy-7,8dihydrocodeinone, OXY) and codeine (7,8-didehydro-4,5-epoxy-3-methoxy-17methylmorphinan-6-ol monohydrate, COD) belong to the class of opioid analgesics used for pain relief. The use of opioid analgesics is controversial due to issues like dependence, tolerance, addiction, and abuse [79]. A CoFe_2_O_4_-modified carbon paste electrode was used for the detection of OXY and COD simultaneously in human plasma and urine samples [80]. The enhanced electrocatalytic activity of the modified electrode provided a good platform for oxidation of OXY and COD. The sensor yielded a detection limit of 0.050 μmol L^−1^ for OXY and 0.02 μmol L^−1^ for COD. Clonazepam (CNZ), (5-(2-Chlorophenyl)-1, 3-dihydro-7-nitro-2H-1, 4-benzodiazepin-2-one), also known as Klonopin, belongs to the group of benzodiazepines. CNZ is used for the treatment of seizures, anxiety, insomnia, and amnesia. CNZ is misused criminally and recreationally. It can cause impaired coordination, distress, and coma [81]. A hybrid nanocomposite of cobalt oxyhydroxide nanoflakes and an rGO nanosheet as the support (CoOOH/r-GO) was used for the detection of CNZ [82]. The synthesis scheme of the CoOOH nanoflakes and CoOOH-rGO nanocomposite is given in Figure 2D. The CoOOH/r-GO/SPCE outperformed the r-GO/SPCE, CoOOH/SPCE, and bare SPCE, which is evident from the DPV curve as shown in Figure 2E. A CoOOH/r-GO/SPCE was used to analyze CNZ using the DPV technique with variation in CNZ concentration (Figure 2F), giving a linear relationship between current signal and CNZ concentration. From this, the detection limit was found to be 38 nM and the sensor performed well in the linear range of 0–350 µM. Five illicit drugs, including cocaine, heroin, MDMA, 4-chloro-alpha-pyrrolidinovalerophenone, and ketamine, were detected simultaneously with an SDS-functionalized SPE with square-wave adsorptive stripping voltammetry (SWAdSV) [83]. The SDS adsorbed at the electrode surface provided adsorption sites for illicit drugs and hence improved the electrochemical output. The sensor yielded a detection limit of 0.7 µM, 1.8 µM, 0.9 µM, 1.6 µM, and 1.1 µM for cocaine, heroin, MDMA, Cl-PVP, and ketamine, respectively.

### 3.3. Electrochemical Sensors for Detection of Personal Care Products (PCPs)

PCPs are an important category of CECs. PCPs include a number of chemicals used in cosmetics and daily care products including UV filters, fragrances, plasticizers, and preservatives. PCPs are detected in aquatic environments worldwide [84]. Since PCPs are mostly used on the human epidermis, they have a higher chance to spread through the atmosphere. PCPs may bioaccumulate in aquatic organisms, and cause endocrine effects, antibiotic resistance, and reproductive malfunctions such as reduction in semen quality or changes in normal development of male genitals [31].

Parabens (alkyl-p-hydroxybenzoates) are commonly used antimicrobial preservatives used in cosmetics, toiletries, pharmaceuticals, and food to increase the shelf life of products. Benzyl, butyl, ethyl, isobutyl, isopropyl, methyl, and propyl parabens are common parabens in use. Of these, methyl and propyl parabens are commonly used in cosmetics. Methyl parabens can cause allergic contact dermatitis and breast cancer [85]. Two electrochemical sensors based on gold nanoparticles decorated with activated carbon-modified pencil graphite electrodes (AuNPs@AC/PGEs) and gold nanoparticles decorated with graphene oxide-modified pencil graphite electrodes (AuNPs@GO/PGEs) were developed, and their ability to sense methyl parabens in cosmetic samples were compared [86]. The AuNPs@GO/PGE showed a better performance by giving a low detection limit of 2.02 μM against the 2.17 μM given by the AuNPs@AC/PGE. Zinc oxide nanoparticles of different morphologies—ZnO nanowires (ZA 8), nanocuboids (ZA 10) and nanospheres (ZA 12)—were synthesized by varying the pH, and their sensitivity towards the detection of methyl parabens were compared [87]. The GCE/ZA 8 exhibited a better performance than both the GCE/ZA 10 and GCE/ZA12 towards methyl paraben detection. The GCE/ZA 8 demonstrated a linear range of 0.02–0.12 mM and a detection limit of 7.25 µM. The electrode also proved to be an attractive candidate for antifouling applications.

Dihydroxybenzenes have two hydroxyl groups (–OH) on the benzene ring. The three structural isomers of dihydroxybenzene are 1,2-dihydroxybenzene (catechol, CC), 1,3-dihydroxybenzene (resorcinol, RS), and 1,4-dihydroxybenzene (hydroquinone, HQ). They are widely used in cosmetics, especially in whitening creams and hair dyes. They are associated with carcinogenesis, allergies, and DNA damage [88]. Nitrogen-doped nickel carbide spheres (N-NiCSs) were synthesized by varying the type of surfactant, surfactant-to-Ni molar ratio, reaction temperature, and reaction time [89]. The scheme for the synthesis of N-NiCS with its fabrication over a GCE and electrochemical detection of CC, RS, and HQ is shown in Figure 3A. Upon comparison of the electrochemical activity of a bare GCE, the N-doped nickel oxide spheres/GCE (N-NiOS/GCE) and the N-NiCS/GCE, the N-NiCS/GCE appeared to have better sensitivity for the detection of HQ, CC, and RS from the DPV analysis as shown in Figure 3B. The DPV analysis with an increase in concentration of the three dihydroxybenzene isomers exhibited an increase in the peak current value (Figure 3C). A linear relationship was established between the concentration of the HQ, CC, and RS and current value. The detection limits of HQ, CC, and RS were inferred from the graph as 0.00152 µM, 0.015 µM, and 0.24 µM, respectively, and linear range of detection as 0.005–100 µM, 0.05–200 µM, and 5–500 µM for HQ, CC, and RS, respectively.

Tert-butylhydroquinone (TBHQ) is a synthetic phenolic antioxidant used as an additive to prevent the oxidation of food, cosmetic, oils, and fats during storage and processing. TBHQ overuse can lead to stomach, DNA, and liver damage. GCE modified with MnO_2_ electrodeposited on electrochemically reduced graphene oxide (MnO_2_/ERGO/GCE) was developed for the detection of TBHQ [90]. The sensor has demonstrated a linear range of 1–50 µM and 100–300 µM and detection limit of 0.8 µM. Vertically ordered mesoporous silica films were fabricated on graphene (VMSF/ErGO/GCE) for the electrochemical detection of TBHQ in cosmetics and edible oils [91]. GO was reduced by cathodic reduction and vertically ordered mesoporous silica films were grown over ErGO by an electrochemical assisted self-assembly (EASA) method, as shown in Figure 3D. The electrochemical performance of the VMSF/ErGO/GCE was better than the bare GCE, ErGO/GCE, and GO/GCE, which was evident from the CV curves (Figure 3E). Using DPV technique, the VMSF/ErGO/GCE could detect TBHQ at a detection limit of 0.23 nM and linear range of 0.001–0.5 and 0.5–120 µM using the DPV technique (Figure 3F). Another electrochemical sensor based on ZnO/ZnNi_2_O_4_ with porous carbon@covalent-organic framework (ZnO/ZnNi_2_O_4_ @porous carbon@COF_TM_) was constructed for the detection of paracetamol and TBHQ [32]. The formed core-shell ZnO/ZnNi_2_O_4_ @porous carbon@COF_TM_ nanocomposite had an active ZnO/ZnNi_2_O_4_@porous carbon as the core and porous N, O-rich COF_TM_ as the shell. The detection limit for paracetamol and TBHQ was 12 nM and 15.95 nM, respectively.

Ultraviolet filters are considered as CECs. They are found in PCPs and industrial products like plastics, paints, rubber, etc., which require protection from photodegradation. With rising awareness of the skin cancer and photoaging associated with sun radiation, the use of sun protection factor (SPF) has become inevitable. However, the organic ultra-violet filter (UVF) tends to bioaccumulate and thus gets transferred to the offspring. It also shows damaging effects on the Central Nervous System and reproductive organs [92]. Oxybenzone and octocrylene are two commonly used UV filters. 2-hydroxy-4-methoxybenzophenone (oxybenzone/benzophenone-3, BP3) and octocrylene (2-Ethylhexyl 2-cyano-3,3-diphenylprop-2-enoate, OC) were simultaneously detected using an SPCE [93]. The electrochemical sensor could sense BP3 at a detection limit of 1.9 µmolL^−1^ and OC at a limit of detection of 4.1 µmolL^−1^.

Triclosan (2,4,40-trichloro-20-hydroxydiphenyl ether, TCS) is an antibacterial agent used widely in toothpaste, soaps, detergents, deodorants, sanitizers, and cleansers [94]. TCS can prevent fatty acid formation and damage chlorophyll in algae. On exposure to light, TCS breaks down, releasing chlorophenols and low-chlorinated dioxins, which are poisonous [95]. A ternary nanocomposite of an rGO-modified porous Cu–benzene tricarboxylic acid metal organic framework (Cu–BTC MOF) decorated on a NiCo bimetallic nanoparticle-modified GCE was fabricated for the detection of TCS [96]. The high electrical conductivity of rGO, greater surface area of Cu–BTC MOF, and the electrocatalytic nature of NiCo bimetallic nanoparticles resulted in a low detection limit of 0.23 × 10^−12^ M and linear range of 49 × 10^−6^ M to 0.39 × 10^−12^ M.

Vanillin (4-hydroxy-3-methoxybenzaldehyde) is an aromatic compound used as a fragrance constituent in cosmetics and as a food additive in chocolates, ice creams, and beverages. Synthetic vanilla, if consumed in unchecked quantities, can cause headaches, nausea, and may affect kidney functions [97]. Manganese dioxide nanowire functionalized with a reduced graphene oxide-modified GCE (MnO_2_NWs-rGO/GCE) was effective for the detection of vanillin [98]. The unique electrochemical properties of r-GO and MnO_2_ NWs resulted in a significant improvement of the oxidation peak of vanillin. The sensor could yield a detection limit of 6 nM and linear ranges of 0.01–20 μM and 20–100 μM from an SDLSV curve.

### 3.4. Electrochemical Sensors for Detection of Endocrine-Disrupting Chemicals (EDCs)

According to the WHO, Endocrine-Disrupting Chemicals (EDCs) are exogenous substances that can interfere with human and animal endocrine systems and thus adversely affect organisms and even their progeny [99]. Bisphenols (BPs), polychlorinated biphenyls (PCBs), phthalate esters, alkylphenols, and natural and synthetic estrogens are common EDCs, as reported. Even minute concentration of EDCs in water can have acute health impacts on living organisms. Reproductive issues, neurological damages, cardiovascular diseases, diabetes, and many other diseases may be caused by exposure to EDCs [100].

Bisphenol A (4,4’-(propane-2,2-diyl) diphenol, BPA) is a component widely used in plastics, baby bottles, storage containers, and epoxy coatings in metal cans. BPA belongs to the class of EDCs and can lead to reproductive issues, infertility, early puberty, cardiovascular diseases, cancer, and neurological and metabolic diseases. A nanocomposite of MAX phase material and Mo_2_Ti_2_AlC_3_ with MWCNT (Mo_2_Ti_2_AlC_3_/MWCNT) was synthesized for sensing BPA in milk packs, plastic bottles, and metal cans [101]. The schematic representation of the detection mechanism is given in Figure 4A. The synergistic effects of Mo_2_Ti_2_AlC_3_ and MWCNT improved the electrocatalytic activity of the electrode compared to the Mo_2_Ti_2_AlC_3_/GCE and MWCNT/GCE, as depicted by the CV curves in Figure 4B. With the use of the DPV technique, the Mo_2_Ti_2_AlC_3_/MWCNT/GCE was used for sensing BPA by increasing the concentration of BPA as shown in Figure 4C. Then, a linear relationship was obtained for the concentration of BPA and the current value, from which a limit of detection of 2.7 nM and linear range of 0.01–8.50 μM were figured out. Recently, another electrochemical sensor using a composite of 2-dimensional layered titanium carbide (Ti_3_C_2_T_x_) MXene with a V_2_O_5_-modified GCE was fabricated for the detection of BPA [102]. The DPV technique exhibited a detection limit of 87 nM and linear range of 414 nM–31.2 μM for BPA sensing. Bisphenol S (Bis(4-hydroxyphenyl) sulfone, BPS) is similar in structure to BPA, except for the sulfone (SO_2_) group present in the central linker instead of the dimethylmethylene group (C(CH_3_)_2_) as in the case of BPA. BPS is often used as a substitute for BPA. BPS also shows adverse effects on reproductive health, heart, and neurobehaviors. Due to their similarities in structure and health effects, BPA and BPS need to be determined simultaneously. A highly porous covalent organic framework, CTpPa-2-modified GCE, was constructed for the simultaneous detection of BPA and BPS in bottle samples [33]. The sensor could detect BPA and BPS at a detection limit of 0.02 µM and 0.09 µM and linear range of 0.1–50 µM and 0.5–50 µM, respectively. Dimethyl bisphenol A (2,2-bis(4-hydroxy-3-toluene), DM-BPA) is another EDC, and has similar adverse effects. Simultaneous detection of BPA and DM-BPA was reported with an SPCE modified with a composite of Pt nanoparticles modified on single-walled carbon nanotubes (Pt@ SWCNT), MXene (Ti_3_C_2_), and graphene oxide (GO) (Pt@SWCNTs-Ti_3_C_2_-rGO/SPCE) [103]. The detection limit for BPA and DM-BPA obtained with the DPV technique were 2.8 nmolL^−1^ and 3 nmolL^−1^, respectively. Tetrabromobisphenol A (4,4′-(Propane-2,2-diyl) bis(2,6-dibromophenol), TBBPA) is another EDC. A conductive composite of carboxylic carbon nanotubes and a cobalt imidazole framework (CNTs@ZIF-67) modified on an acetylene black electrode (CNTs@ZIF-67/ABE) was developed for TBBPA detection, and the composite showed an excellent adsorption capacity for TBBPA detection [104]. This resulted in a low detection limit of 4.23 nmolL^−1^ and linear range of 0.01–1.5 µmolL^−1^.

Bisphenol F (4,4′-Methylenediphenol, BPF), a structural analogue of BPA, is used as an alternative for BPA in the market. BPF has the potency to cause fertility defects and adverse effects on brain development in mammals, and depletes glutathione levels in humans, resulting in an increased oxidative stress to raise cancer incidences [105]. An electrochemical sensor with a carbon paste electrode modified with zinc oxide-reduced graphene oxide and cetyltrimethylammonium bromide (ZnO/G/CTAB/MPCE) was used to detect BPF in water samples, human body fluids, and canned drinks [106]. BPF was recognized with the DPV technique, and it yielded a detection limit of 0.06 µM and linear range of 0.5–10 µM.

BPSIP (4-(4-isopropoxy-benzenesulfonyl)-phenol) was introduced as an alternative dye developer to BPS. BPSIP also has endocrine-disrupting effects, such as estrogenicity and antiandrogenicity [34]. BPSIP sensing in river water samples was implemented using a nanocomposite of graphitic carbon nitride and copper coordinated dithiooxamide metal organic framework modified on a GCE (g-C_3_N_4_/Cu-DTO MOF/GCE) [107]. A detection limit of 0.02 μM and sensitivity of 0.5675 μAμM^−1^cm^−2^ was achieved using the sensor with the DPV technique. Recently, another electrochemical sensor based on a graphene-modified SPCE (g-C_3_N_4_@GN/SPCE) was developed for the detection of BPSIP [108]. The g-C_3_N_4_@GN/SPCE showed a good electrochemical response to BPSIP oxidation due to the synergistic effect of g-C_3_N_4_ and the GN/SPCE. The sensor offered a detection limit of 0.02 ± 0.01 μM and linear ranges of 1–100 μM and 100–1000 μM.

o-phenylphenol (OPP) and butylparaben (BP) are Endocrine-Disrupting Chemicals. Mn_2_O_3_ samples with various morphologies, including spherical, dumbbell-like, cubic, and ellipsoidal (S-Mn_2_O_3_, D-Mn_2_O_3_, C-Mn_2_O_3_, E-Mn_2_O_3_), were synthesized and their activity towards OPP and BP sensing were evaluated [109]. Of these, E-Mn_2_O_3_ showed a better performance. In order to improve the performance of the sensor, a GO-wrapped E-Mn_2_O_3_-modified GCE was developed (E-Mn_2_O_3_@GO/GCE), which yielded a detection limit of 0.63 nM and 0.88 nM for OPP and BP, respectively.

17α-Ethinylestradiol (EE2) is a synthetic hormone used widely as a contraceptive, for the treatment of menopausal and postmenopausal syndrome, and breast cancer in postmenopausal women. However, a higher amount of EE2 intake can cause sexual abnormalities, decrease fertility, and cause cancer [110]. Carbon black-supported palladium nanoparticles (CB/Pd NPs) were synthesized for the electrochemical detection of EE2 [111]. The synergistic effect of CB and Pd NPs enhanced the electrooxidation of EE2. The detection limit attained with the CB/PdNPs/GCE was 81 nmolL^−1^ and linear range of 0.5–119 µmolL^−1^. The compound 17 β-estradiol (17 β-E2) is a natural steroid hormone existing in mammals. Side effects of 17 β-E2 include endocrine disorders, cancer, obesity, diabetes, and neurological disorders [112]. Wrinkled mesoporous carbon synthesized from wrinkled silicon nanoparticle as a sacrificial template with a modified GCE (*w*MC_0.67_/GCE) was used for the detection of 17 β-E2 [113]. The step-by-step procedure for the synthesis of the *w*MC/GCE and the mechanism for the detection of 17 β-E2 is given in Figure 4D. The *w*MC_0.67_/GCE was demonstrated to have a better electrocatalytic activity and electron transfer compared to a bare GCE and other modified electrodes, as shown in the CV curves of Figure 4E. With the increase in concentration of 17 β-E2 added, the peak current value showed an increase in Figure 4F. From the linear relationship drawn between the concentration of 17 β-E2 and the current value, a detection limit of 8.3 nM and linear ranges of 0.05–10 and 10–80 µM were accomplished using the sensor, by increasing the concentration of 17 β-E2 in real samples such as milk and river water. Diethylstilbestrol (DES) is another hormone that has similar side effects as 17 β-E2. An electrochemical sensor for the simultaneous analysis of 17 β-E2 and DES using Fe_3_O_4_-doped nanoporous carbon (Fe_3_O_4_-NC) synthesized by the carbonization of Fe-porous coordination polymer (Fe-PCP) was developed [114]. The strong adsorptive and catalytic performance of Fe_3_O_4_-NC granted a detection limit of 4.6 nmolL^−1^ and 4.9 nmolL^−1^, and linear range of 0.01–12 µmolL^−1^ and 0.01–20 µmolL^−1^ for sensing DES and 17 β-E2, respectively.

### 3.5. Electrochemical Sensors for Detection of Newly Registered Pesticides (NRPs)

Pesticides are a group of chemicals and microorganisms for the eradication of insects, fungi, weeds, and bacteria [115]. Different classes of pesticides include organophosphates, organochlorines, carbamates, pyrethrin, neonicotinoids, sulfonylureas, and triazines. Newly registered pesticides are new pesticide chemicals or a different use of an existing chemical as pesticide. In order to cater to the demands of food production worldwide, the agriculture sector is largely dependent on pesticides. Even though pesticides increase the crop, vegetable, and fruit yields by several folds, the havoc they cause is a matter of serious concern. Soil and water contamination, bioaccumulation, carcinogenic effects, damage to the immune system, reproductive system, nervous system, and respiratory system are the dangerous impacts of the use of pesticides [116].

Neonicotinoids are a group of pesticides which are highly effective against insects. Imidacloprid (IDP), thiamethoxam (TMX), and dinotefuran (DNF) are common neonicotinoid pesticides. IDP (1-(6-chloro-3-pyridymethyl)-N-nitroimidazolidin-2-ylideneamine) is a neonicotinoid pesticide. IDP binds irreversibly to the nicotinic acetylcholine receptors which are responsible for the functioning of the central nervous system. This hinders neural transmission, eventually leading to the paralysis and then the death of the insect [117]. IDP may also cause liver failure, cancer, and affect neural tissue development in infants [118]. An electrochemical sensor for the detection of IDP using Ag nanoparticles deposited on mesoporous carbon and a naturally extracted hematite ore (Ag@Meso-C/Hematite Ore) nanocomposite resulted in a detection limit of 0.257 μM [35]. An organic/inorganic composite of f-MWCNT/EDTA integrated electrochemical sensor was used for the detection of IDP [36]. This electrochemical sensor provided a detection limit of 3.1 × 10^−3^ pM. Tungsten sulfide (WS_2_) nanosheets were fabricated for the detection of IDP in water and soil samples [119]. Fast electron transfer on WS_2_ nanosheets resulted in electrochemical reduction of the aromatic nitro group in IDP at a detection limit of 0.28 µM and linear range of 10–90 µM. A hydrothermally synthesized Fe-rich FeCoNi-MOF in-situ modified nickel foam working electrode was utilized for sensing IDP [120], as shown in Figure 5A. An optimized ratio of Fe:Co:Ni (5:1:1) exhibited a better electrocatalytic activity for the detection of IDP in Figure 5B. The exposed rich active sites in Fe-rich FeCoNi-MOF resulted in a low detection limit of 0.04 pmol/L and linear range of 1–1.2 × 10^8^ pmol/L, using the DPV technique as shown in Figure 5C.

TMX ((3-[(2-chloro-5-thiazolyl)methyl]tetrahydro-5-methyl-N-nitro-4H-1,3,5-oxadiazine-4-imine) is a neonicotinoid pesticide. Food safety authorities in European countries have restricted the range of TMX in agronomy products as 0.02 mg/kg to 5.0 mg/kg [121]. A composite of hydrothermally synthesized cobalt oxide in graphitic carbon nitride (Co_3_O_4_@g-C_3_N_4_NC) was used for sensing TMX [122], as shown in Figure 5D. Due to the large electroactive surface area and faster electron transfer offered by the composite, the Co_3_O_4_@g-C_3_N_4_/SPCE showed a better electrocatalytic performance compared to the Co_3_O_4_/SPCE, g-C_3_N_4_/SPCE and bare SPCE, as shown by CV analysis in Figure 5E. DPV techniques showed a detection limit of 4.9 nM and linear range of 0.01–420 µM in Figure 5F. A Fe_2_O_3_@g-C_3_N_4_@Schiff base-modified GCE synthesized by a calcination method was used for sensing TMX [123]. The Fe_2_O_3_@g-C_3_N_4_@MSB/GCE provided a detection limit of 0.137 µM and linear range of 0.01–200 µM for sensing TMX in real samples, like potato, rice, and river water. Another electrochemical sensor was fabricated with a three-dimensional nitrogen-doped macro-meso-microporous carbon composite derived from nitrogen-doped Cu-MOF, using PVP (N/Cu–HPC) for the simultaneous detection of neonicotinoids, such as IDP, TMX, and DNF [37]. Mass and charge transport between neonicotinoid analyte molecules and Cu nanoparticles and carbon atoms improved the electrocatalytic performance of the N/Cu–HPC/GCE towards the detection of IDP, TMX, and DNF. The N/Cu–HPC/GCE provided a detection limit of 0.026 μM for IDP, 0.062 μM for TMX, and 0.01 μM for DNF, and linear ranges of 0.5–60 μM for both IDP and DNF, and 1–60 μM for TMX. Another electrochemical sensor using a MOF-derived N-doped octahedral NiCu nanoporous carbon composite-modified GCE (N/NiCu@C/GCE) was developed for the detection of neonicotinoids IDP, TMX, and DNF [124]. The porous carbon structure and NiCu nanoalloy improved the diffusion between the electrolyte and active site, enhancing the electrocatalytic performance of the N/NiCu@C/GCE. Using the DPV technique, the detection limit and linear ranges for sensing IDP, TMX, and DNF were found out to be 0.017, 0.007, and 0.001 μM, and 0.5–60, 1–60, and 0.5–60 μM, respectively.

### 3.6. Electrochemical Sensors for Detection of Disinfection By-Products (DBPs)

Drinking water disinfection to ensure safe drinking water was one of the greatest achievements of the past century. However, the by-products of the process are highly toxic. Disinfection By-Products (DBPs) are formed when disinfectants like chlorine, ozone, chlorine dioxide, or chloramines, react with naturally occurring organic matter, man-made contaminants, bromide, and iodide during drinking water production [125]. DBPs are associated with genotoxicity and carcinogenicity [126]. Chlorates, chlorites, bromates, trihalomethanes (THMs), and haloacetic acids (HAAs) are common DBPs found in drinking water.

Chlorine and its oxides are the most common and cost-effective option for drinking water disinfection. Use of ClO_2_^−^ may cause acquired methemoglobinemia, which has a life-threatening potential by reducing the oxygen capacity of hemoglobin [127]. The WHO and EPA regulated that the concentration of ClO_2_^−^ should not exceed above 0.2 mg/L [128]. Fe_3_O_4_ nanoparticles synthesized by surfactant-assisted hydrolysis of optimized amounts of Fe^2+^ and Fe^3+^ ions modified on a CPE was used for the electrochemical sensing of ClO_2_^−^ [129]. The sensor could detect ClO_2_^−^ ions in an aqueous medium. The practical applicability of the sensor was tested in tap water and bottled water samples. The sensor showed a detection limit of 8.6 nM. Another electrochemical sensor with a carbon black-modified SPCE was used as the working electrode (CB-SPCE) for the detection of ClO_2_^−^ [130]. The sensor could detect ClO_2_^−^ at a limit of detection of 0.01 ppm and linear range of 0.05–20 ppm. The sensor was found to be effective in sensing ClO_2_^−^ in swimming pool water samples.

Bromate (BrO_3_^−^) is also a by-product of the ozonation and chlorination disinfection process. Potassium bromate (KBrO_3_) is used as an oxidizing agent in the baking industry. Bromate is classified as a B2 carcinogen. The US EPA and WHO have recommended 10 µg/L as the maximum acceptable concentration of BrO_3_^−^ in drinking water [129]. For the detection of BrO_3_^−^, a nanocomposite of electrochemically polymerized poly(aniline-co-o-aminophenol), ERGO, and Pd (PANOA/ERGO/Pd) with a caterpillar-like structure modified GCE was integrated [131]. The sensor produced a detection limit of 1 µM and linear range of 4–840 µM for BrO_3_^−^ detection. The sensor performed well in drinking water and river water samples. A lamellar MXene, Ti_3_C_2_T_x_-modified GCE was fabricated for BrO_3_^−^ in drinking water samples [132]. Figure 6A shows the fabrication of the working electrode by GCE and the electrochemical detection of BrO_3_^−^. The electrochemical activity of MXene for BrO_3_^−^ detection was confirmed with the CV comparison of a bare GCE, the GCE/Graphene, and the GCE/MXene as shown in Figure 6B. Mxene as a sensing platform electrocatalytically augmented the reduction of BrO_3_^−^, which resulted in a detection limit of 41 nM and linear range of 50 nM–5 µM. The detection limit and linear ranges for BrO_3_^−^ detection was obtained from the DPV curve of increasing concentration of BrO_3_^−^, as given in Figure 6C. The practical applicability of the MXene sensor was also tested in domestic tap water.

Trichloroacetic acid (TCAA) is a kind of haloacetic acid (HAA). TCAA is associated with carcinogenicity, reproductive defects, and mutagenicity [133]. Combining the redox activity of iron phthalocyanine (PcFe) and high surface area and absorbability of ZIF-8, an efficient electrochemical sensor (PcFe@ZIF-8) for TCAA sensing was developed [134]. The sensor exhibited a detection limit of 1.89 nM. The sensor also showed good performance in real samples, such as tap water and swimming pool water samples. Haloacetamides (HAM) are a class of DBPs. Although the concentration of HAMs are lower than THMs and HAAs, their cytotoxic and genotoxic effects are twofold [135]. Trichloroacetamide (TCAM), an HAM, was electrochemically detected with a heterostructure of a triangular Ag nanoprism@MoS_2_ nanosheet (AgNPR@MoS_2_/GCE) [136]. The schematics of the electrochemical detection of TCAM on the AgNPR@MoS_2_/GCE are given in Figure 6D. The supremacy of the AgNPR@MoS_2_/GCE for electrochemical detection of TCAM can be clearly seen from SWV curves of the bare GCE, AgNPR/GCE, MoS_2_/GCE, and AgNPR@MoS_2_/GCE. The reduction peak current was better compared to other electrodes, as shown in Figure 6E. Quantitative detection of TCAM was conducted using the DPV technique. With the increase in concentration of the added TCAM analyte, the peak current value also increased, as shown in Figure 6F. Due to the dechlorination reaction of TCAM catalyzed by AgNPR and accelerated by the H^+^ absorption by S atoms in MoS_2_ nanosheets, the sensor could provide a detection limit of 0.17 µM and linear ranges of 0.5–10 μM and 10–80 μM. The applicability of the sensor in TCAM detection in drinking water samples was also demonstrated.

### 3.7. Electrochemical Sensors for Detection of per- and Polyfluoroalkyl Substances (PFAs)

PFAs are aliphatic substances containing the moiety C*_n_*F_2*n*+1_ within the structure, where *n* ≥ 1. This group also includes substances where a perfluorocarbon chain is connected with functional groups on both ends, aromatic substances that have perfluoroalkyl moieties on the side chains, and fluorinated cycloaliphatic substances [137]. PFAs are known as “forever chemicals” as they resist breakdown under natural conditions and have a half-life of >92 years in water. These compounds are used in the synthesis of packaging products, carpets, firefighting foams, paints, semiconductors, carpets, etc. The fluorinated regions in PFAs offer unique properties, such as thermal stability due to the presence of strong C-F bonds, stability against acids, bases, oxidizing and reducing agents, and oil and water repellence [138]. The toxic effects of PFAs include immunological, cardiovascular, reproductive, developmental, and liver effects [38]. Perfluorooctanoic acid (PFOA), perfluorooctane sulfonate (PFOS), GenX, fluorotelomer alcohols (FTOHs), and polyfluoroalkyl phosphates are common PFAs. The Heads of the Environmental Protection Agencies for Australia and New Zealand (HEPA) set the PFA guideline values as 0.00023 μg/kg for PFOS and 19 μg/kg for PFOA in aquatic environments [139]. An electrochemical sensor using Hf-doped WO_3_ was synthesized by a hydrothermal method modified on a CPE for the detection of PFOA [140]. The higher surface area of the modifier improved the electrocatalytic activity. The limit of detection and linear range were found to be 1.83 × 10^−8^ M and 7.0 × 10^−8^ M to 3.0 × 10^−4^ M, respectively, using the SWV technique. A nanocomposite of WS_2_-MWCNT was also used for the electrochemical detection of PFOA as well as for a supercapacitor application [141]. The synergistic effects of MWCNT and WS_2_ enhanced the electronic conductivity and sensitivity for the detection of PFOA, where the sensor yielded a limit of detection of 2.404 pM. Table 3 summarizes the comparison of the merits of electrochemical sensors for the detection of different classes of CECs, as discussed in the review.

## 4. Conclusions and Future Perspectives

CECs are so toxic that the presence of very minute quantities can have adverse effects on the health of living beings. Since the wastewater treatment fails to remove CECs, they continue to exist in drinking water, entering the body of living organisms, bioaccumulating, and causing hazardous effects. Electrochemical sensors help in monitoring these CECs in real samples both qualitatively and quantitively, and thus ensure safety in food and drinking water. Therefore, a number of electrochemical sensors have been developed for CEC detection in real samples. Electrochemical sensors have been developed for the detection of pharmaceuticals, PCPs, illicit drugs, EDCs, pesticides, and DBPs.

Already known CECs should be detected, and if found above the regulatory limits of international agencies, like the WHO and EPA, proper action must be taken to prevent their noxious effects. More efficient electrochemical sensors must be developed for the simultaneous detection of CECs which can be present together, because combinations of some CECs can be more toxic than their individual presence. Such combinations must be monitored with a single electrochemical sensor. Also, there are a number of CECs for which there are no reports of electrochemical sensors available, which need to be properly identified. Sensors should be developed for the sensitive and selective detection of those CECs. Although the number of chemicals classified under CEC is increasing, there are a number of chemicals which are still left unidentified as CECs as their harmful effects are unknown. Such toxic chemicals need to be screened and their use needs to be regulated.

## Figures and Tables

**Figure 1 molecules-28-07916-f001:**
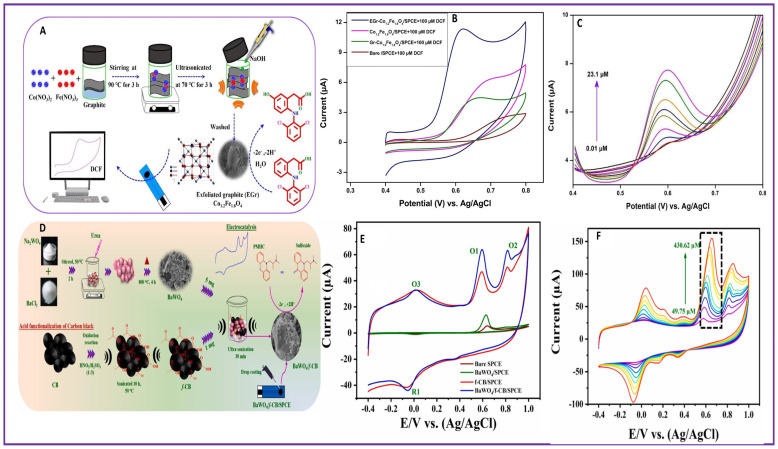
Scheme of fabrication (**A**) EGr-Co_1.2_Fe_1.8_O_4_/SPCE, (**D**) BaWO_4_/*f*-CB/SPCE; Comparison of CVs of different electrodes (indicated by different colors) for detection of (**B**) DIC, (**E**) PMHC; (**C**) DPV of EGr-Co_1.2_Fe_1.8_O_4_/SPCE for detection of DIC at different concentrations (indicated by different colors); (**F**) CV of BaWO_4_/*f*-CB/SPCE for detection of PMHC at different concentrations (indicated by different colors).

**Figure 2 molecules-28-07916-f002:**
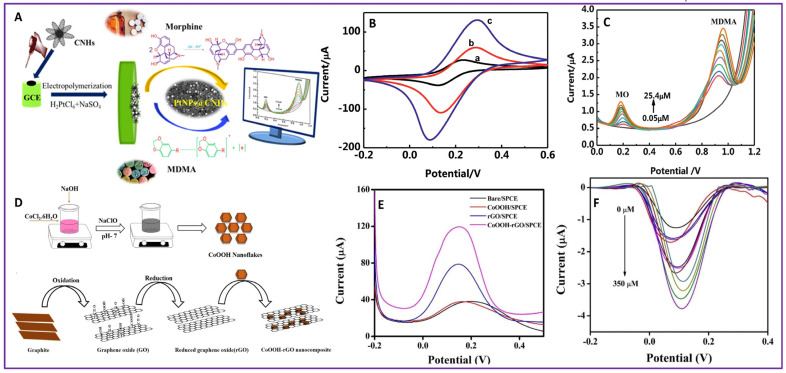
Scheme of fabrication of (**A**) CNH-CHI@PtNPs/GCE, (**D**) CoOOH/r-GO/SPCE; (**B**) Com parison of CVs of different electrodes (indicated by different colors) for detection of MO and MDMA; (**C**) DPV of CN-CHI@PtNPs/GCE at different concentrations(indicated by different colors) of MO and MDMA; (**E**) Comparison of DPVs of different electrodes (indicated by different colors) for detection of CNZ; (**F**) DPV of CoOOH/r-GO/SPCE at different concentrations (indicated by different colors) CNZ.

**Figure 3 molecules-28-07916-f003:**
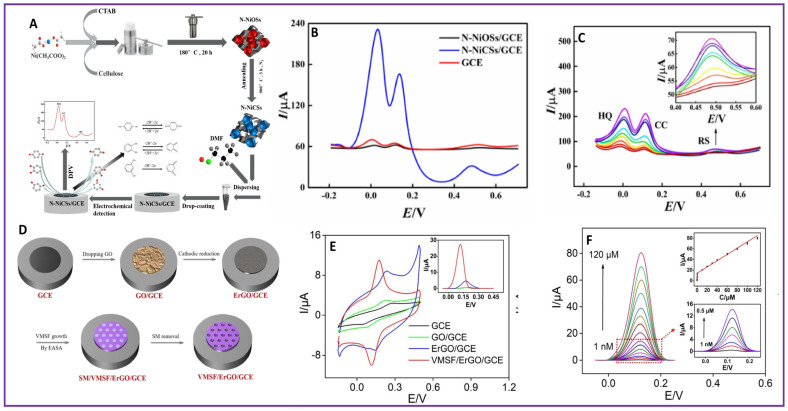
Scheme of fabrication of (**A**) N-NiCS/GCE (**D**) VMSF/ErGO/GCE; (**B**) Comparison of DPVs of different electrodes (indicated by different colors) for detection of HQ, CC, and RS; (**E**) CVs of different electrodes (indicated by different colors) for detection of TBHQ; (**C**) DPV curves of N-NiCS/GCE at different concentrations (indicated by different colors) of HQ, CC and RS; (**F**) VMSF/ErGO/GCE at different concentrations (indicated by different colors) of TBHQ.

**Figure 4 molecules-28-07916-f004:**
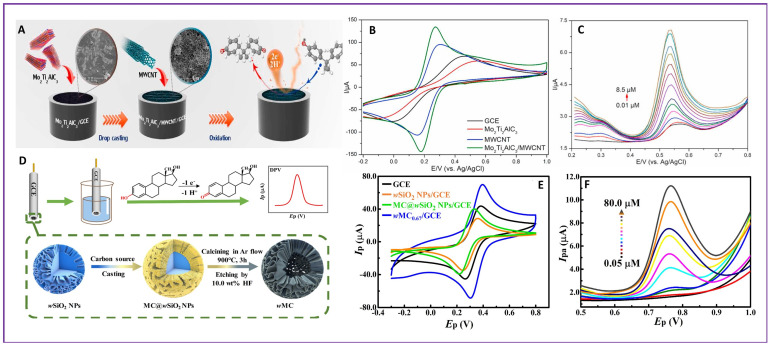
Scheme of fabrication of (**A**) Mo_2_Ti_2_AlC_3_/MWCNT/GCE (**D**) *w*MC/GCE; Comparison of CVs of different electrodes ((indicated by different colors) for detection of (**B**) BPA, (**E**) 17 β-E2; (**C**) DPVs of Mo_2_Ti_2_AlC_3_/MWCNT/GCE at different concentrations (indicated by different colors) of BPA; (**F**) *w*MC/GCE at different concentrations (indicated by different colors) of 17 β-E2.

**Figure 5 molecules-28-07916-f005:**
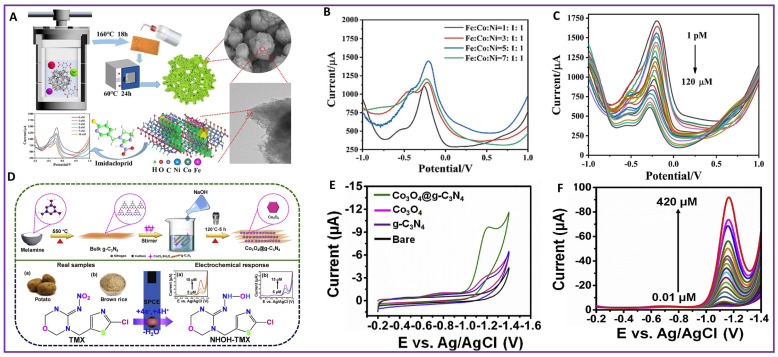
Scheme of fabrication of (**A**) FeCoNi-MOFs (**D**) Co_3_O_4_@g-C_3_N_4_NC; (**B**) Comparison of DPV of different electrodes of FeCoNi-MOFs for detection of IDP; (**C**) DPV curve of Nickel foam/Fe-rich FeCoNi-MOF at different concentrations (indicated by different colors) of IDP; (**E**) CVs of different electrodes (indicated by different colors) for detection of TMX; (**F**) Co_3_O_4_@g-C_3_N_4_/SPCE at different concentrations (indicated by different colors) of TMX.

**Figure 6 molecules-28-07916-f006:**
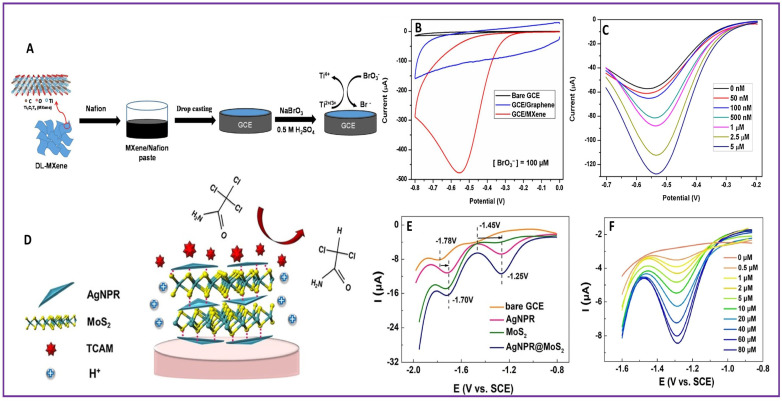
Scheme of electrochemical detection of (**A**) BrO_3_^−^ at Ti_3_C_2_T_x_/GCE (**D**) TCAM at AgNPR@MoS_2_/GCE; Comparison of (**B**) CVs of different electrodes(indicated by different colors) for detection of BrO_3_^−^ (**E**) SWVs of different electrodes(indicated by different colors) for detection of TCAM; (**C**) DPV of Ti_3_C_2_T_x_/GCE at different concentrations(indicated by different colors) of BrO_3_^−^; (**F**) SWV of AgNPR@MoS_2_/GCE at different concentrations(indicated by different colors) of TCAM.

**Table 2 molecules-28-07916-t002:** Comparison of electrochemical sensors, LC–MS/MS techniques, and optical sensors.

Parameters	Electrochemical Sensor	LC–MS/MS	Optical Sensor
Working principle	Converts information from electrochemical reaction between an electrode and analyte into an applicable qualitative or quantitative signal	LC separates the sample components and then introduces them to the mass spectrometer, where the charged ions are created and detected	Detects either wavelength, frequency, or polarization of light and converts it into an electric signal, where the presence of an analyte causes change in absorbance, luminescence, or refractive index of the material resulting in a change in the output signal
Set up	Potentiostat stand, electrochemical cell, and a computer with data acquisition and analysis software	Autosampler, degasser, isocratic/binary quarternary pump, column thermostat connected to tandem mass spectrometer with an ion source controlled with a computer with data acquisition and analysis software, N_2_ gas generator, and a vacuum pump	Light source, adjustable monochromator, sample holder, and a detector with a computer with software for data acquisition and analysis
Cost	Low	Expensive	Cost effective
Operational simplicity	Simple	Difficult	Often simple
Analysis time	Low (in range of seconds)	High (in range of minutes)	Low (in range of seconds)
Selectivity	Fair	High	Good
Data interpretation	Easy	Advanced	Advanced
Sample preparation	Easy	Multi-step	Nil
Response curve	Nernstian (potentiometric/linear)	LC–MS chromatogram	Sigmoidal
Instrument size	Compact	Bulky	Bulky
Portability	Often portable	Non-portable	Often portable
Consumption of organic solvents	Low	High	High

**Table 3 molecules-28-07916-t003:** Comparison of merit of electrochemical sensors of different classes of CECs.

Sl. No.	CEC Class	Analyte	Modified Electrode	LOD	Linear Range	Sensitivity	Ref.
1.	Pharmaceuticals	PA	VFe_2_O_4_/GCE	8.20 nM	0.05–12.3µM	Not reported	[59]
2.	Pharmaceuticals	PA DIC ORP	COOH-CNTs/ZnO/NH_2_-CNTs/GCE	46.8 fM 78 fM 60 fM	25 pM–0.5 µM 75 pM–1.5 µM 75 pM–1.5 µM	Not reported	[60]
3.	Pharmaceuticals	DIC	EGr-Co_1.2_Fe_1.8_O_4_/SPCE	1 nM	0.01–23.1 µM	1.059 µA µM^−1^cm^−2^	[61]
4.	Pharmaceuticals	DS	Oxidized g-C_3_N_4_/Cu–Al LDH/GCE	0.38 μM	0.5–60 μM	0.00646 μA μM^−1^	[62]
5.	Pharmaceuticals	IBU	Cu_3_TeO_6_/GCE	0.017 µM	0.02–5 μM and 9–246 μM	5.97 µA µM^−1^cm^−2^	[63]
6.	Pharmaceuticals	NZ	Sg–C_3_N_4_/CuWO_4_	3 nM	0.005 μM–877 μM	1.24 μA μM^−1^cm^−2^	[64]
7.	Pharmaceuticals	NFX	CaCuSi_4_O_10_/GCE	0.0046 μM	0.01–0.55 and 0.55–82.1 µM	Not reported	[65]
8.	Pharmaceuticals	NAP SUM	ZnO/NiO/Fe_3_O_4_/MWCNTs	3 nM 2 nM	4.00 nM to 350.00 μM 6.00 nM to 380.00 μM	Not reported	[66]
9.	Pharmaceuticals	SUM NAP	P-L CuO: Tb^3+^ NS/CPE	3.3 nM 2.7 nM	0.01–800 μM 0.01–700 μM	Not reported	[67]
10.	Pharmaceuticals	VOR	AuNPs@GRP/GCE	50 nM	0.1–1.0 and 1.0–6.0 μM	Not reported	[26]
11.	Pharmaceuticals	PMHC	BaWO_4_/f-CB/SPCE	29 nM	0.03–234.74 μM and 274.73–1314.73 μM	826 mM μA^−1^cm^−2^	[68]
12.	Pharmaceuticals	CBZ	GdVO_4_/f-CNF/GCE	0.0018 μM	0.01–157 μM	4.8023 μA μM^−1^cm^−2^	[27]
13.	Illicit Drugs	Cocaine	Oh-Pd^2+^: Co_3_O_4_-C/GCE	1.3 nM	0.01 μM–900.0 μM	Not reported	[75]
14.	Illicit Drugs	Morphine	PDA-f-MWCNT/GCE	0.06 µM	0.075–75.0 μM	Not reported	[76]
15.	Illicit Drugs	Morphine MDMA	CNH-CHI@PtNPs/GCE	0.02 µM 0.018 µM	0.05–25.4µM	Not reported	[78]
16.	Illicit Drugs	MDEA	C-SPE	0.03 μM	2.5 to 30.0 µM	0.569 µA/µmol L^−1^	[30]
17.	Illicit Drugs	OXY COD	CoFe_2_O_4_/C-SPE	0.050 μM 0.02 μM	0.06–38 µM	Not reported	[79]
18.	Illicit Drugs	CNZ	CoOOH/r-GO/SPCE	38 nM	0–350µM	0.054 µA µM^−1^cm^−2^	[82]
19.	Illicit Drugs	Cocaine Heroin MDMA Cl-PVP Ketamine	SDS-SPE	0.7 µM 1.8 µM 0.9 µM 1.6 µM 1.1 µM	1–30 µM 2.5–30 µM 1–30 µM 2.5–30 µM 2.5–30 µM	0.25 µA µM^−1^ 0.07 µA µM^−1^ 0.2 µA µM^−1^ 0.06 µA µM^−1^	[83]
20.	PCPs	Methyl parabens	AuNPs@GO/PGE	2.02 μM	0.030–1 mM	Not reported	[86]
21.	PCPs	Methyl parabens	GCE/ZA 8	7.25 µM	0.02–0.12 mM	Not reported	[87]
22.	PCPs	HQ CC RS	N-NiCS/GCE	0.0015 µM 0.015 µM 0.24 µM	0.005–100 µM 0.05–200 µM 5–500 µM	4.635 μA μM^−1^cm^−2^ 2.069 μA μM^−1^cm^−2^ 0.985 μA μM^−1^cm^−2^	[89]
23.	PCPs	TBHQ	MnO_2_/ERGO/GCE	0.8 µM	1–50 µM and 100–300 µM	Not reported	[90]
24.	PCPs	TBHQ	VMSF/ErGO/GCE	0.23 nM	0.001–0.5 and 0.5–120µM	26.17 μA/μM	[91]
25.	PCPs	TBHQ	ZnO/ZnNi_2_O_4_ @porous carbon@COF_TM_	15.95 nM	47.85 nM–130 μM	18.4 μA μM^−1^cm^−2^	[32]
26.	PCPs	BP3 OC	SPE	1.9 µM 4.1 µM	6–200 μM 11–300 μM	1.4 × 10^−3^ AVmol^−1^ L 1.6 × 10^−3^ AVmol^−1^ L	[93]
27.	PCPs	TCS	rGO/Cu–BTCMOF/NiCo/GCE	0.23 pM	0.39 pM–49 μM	0.196 µA/mM	[96]
28.	PCPs	Vanillin	MnO_2_NWs-rGO/GCE	6 nM	0.01–20 μM and 20–100 μM	Not reported	[97]
29.	EDCs	BPA	Mo_2_Ti_2_AlC_3_/MWCNT/GCE	2.7 nM	0.01–8.50 μM	Not reported	[101]
30.	EDCs	BPA	Ti_3_C_2_T_x_/V_2_O_5_/GCE	87 nM	414 nM–31.2 μM	Not reported	[102]
31.	EDCs	BPA BPS	CTpPa-2/GCE	0.02 µM 0.09 µM	0.1–50 µM 0.5–50 µM	Not reported	[33]
32.	EDCs	BPA DM-BPA	Pt@SWCNTs-Ti_3_C_2_-rGO/SPCE	2.8 nM 3 nM	0.006–7.4 μM	1.941 μA (μmol L^−3^)^−1^ cm^−2^	[103]
33.	EDCs	TBBPA	CNTs@ZIF-67/PFDA/AB	4.23 nM	0.01–1.5 µM	Not reported	[104]
34.	EDCs	BPF	ZnO/G/CTAB/MPCE	0.06 µM	0.5–10 µM	Not reported	[106]
35.	EDCs	BPSIP	g-C_3_N_4_/Cu-DTO MOF/GCE	0.02 μM	0.04–1.10 µM	0.5675 µA µM^−1^cm^−2^	[107]
36.	EDCs	BPSIP	g-C_3_N_4_@GN/SPCE	0.02 ± 0.01 μM	1–100 μM and 100–1000 μM	0.9162 ± 0.0003 µA µM^−1^cm^−2^	[108]
37.	EDCs	OPP BP	E-Mn_2_O_3_@GO/GCE	0.63 nM 0.88 nM	0.002–20 μM 0.003–24 μM	Not reported	[109]
38.	EDCs	EE2	CB/Pd NPs/GCE	81 nM	0.5–119 µM	0.176 μA μmol^−1^cm^−2^	[111]
39.	EDCs	17 β-E2	wMC_0.67_/GCE	8.3 nM	0.05–10 and 10–80 µM	Not reported	[113]
40.	EDCs	DES 17 β-E2	Fe_3_O_4_-NC/GCE	4.6 nM 4.9 nM	0.01–12 µM 0.01–20 µM	Not reported	[114]
41.	NRPs	IDP	Ag@Meso-C/Hematite Ore/GCE	0.257 μM	10.80–195.50 μM	0.8113 µM µA^−1^cm^−2^	[35]
42.	NRPs	IDP	f-MWCNT/EDTA/SPCE	3.1 × 10^−3^ pM	0.001–0.05 nM, 0.001–0.04 μM 0.001 nM–0.04 mM	10.70 μAnM^−1^	[36]
43.	NRPs	IDP	WS_2_/GCE	0.28 μM	10–90 µM	3.98 μA μM^–1^cm^–2^	[119]
44.	NRPs	IDP	Nickel foam/Fe-rich FeCoNi-MOF	0.04 pM	1–1.2 × 10^8^ pM	124 μA pmol/L^−1^cm^−2^	[120]
45.	NRPs	TMX	Co_3_O_4_@g-C_3_N_4_/SPCE	0.0049 µM	0.1–420 µM	12.2136 μA μM^−1^cm^−2^	[122]
46.	NRPs	TMX	Fe_2_O_3_@gC_3_N_4_@MSB/GCE	0.137 µM	0.01–200 µM	Not reported	[123]
47.	NRPs	IDP TMX DNF	N/Cu–HPC/GCE	0.026 μM 0.062 μM 0.01 μM	0.5–60 μM 1–60 μM 0.5–60 μM	Not reported	[37]
48.	NRPs	IDP TMX DNF	N/NiCu@C/GCE	0.017 μM 0.007 μM 0.001 μM	0.5–60 μM 1–60 μM 0.5–60 μM	Not reported	[124]
49.	DBPs	ClO_2_^−^	Fe_3_O_4_/CPE	8.6 nM	1–10 μM, 20–100 μM	Not reported	[129]
50.	DBPs	ClO_2_^−^	CB-SPCE	0.01 ppm	0.05–20 ppm	Not reported	[130]
51.	DBPs	BrO_3_^−^	PANOA/ERGO/Pd/GCE	1 µM	4–840 µM	33.2 nA μM^−1^	[131]
52.	DBPs	BrO_3_^−^	Ti_3_C_2_T_x_/GCE	41 nM	50 nM–5 µM	Not reported	[132]
53.	DBPs	TCAA	PcFe@ZIF-8/GCE	1.89 nM	0.02–1 μM	826 μA/μM	[134]
54.	DBPs	TCAM	AgNPR@MoS_2_/GCE	0.17 µM	0.5–10 μM and 10–80 μM.	Not reported	[135]
55.	PFAs	PFOA	Hf.WO_3_/CPE	1.83 × 10^−8^ M	7.0 × 10^−8^ M to 3.0 × 10^−4^ M	Not reported	[140]
56.	PFAs	PFOA	WS_2_-MWCNT	2.404 pM	10–120 pM	Not reported	[141]

## Data Availability

Data can be available upon request from the authors.

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
