# Peer review of "Recent Advances on Electrochemical Sensors for Detection of Contaminants of Emerging Concern (CECs)"

_molecules, 2023, doi:10.3390/molecules28237916_

Round 1

Reviewer 1 Report

Comments and Suggestions for Authors

For a paper on Electrochemical Sensors for Detection of Contaminants of Emerging Concern, PFOS should not be missing. This will generate more interest. The paper would also benefit by a conclusion with some comments on the “best” electrochemical methods and maybe comment on any commonality between the contaminants and the detection to guide future research. Finally, some comparison between other methods would be useful. The benefit of electrochemical detection is mentioned but not in contrast to a specific spherics. For example, LC-triple quad verses an electrochemical cell – noting the difference and relative benefit. The paper would benefit from some insight into the commonality of the electrochemical methods for emerging contaminants and contrast to other mythology. This is important for a review paper.

Comments on the Quality of English Language

English is fine. Minor editing will help the paper. 

Author Response

For a paper on Electrochemical Sensors for Detection of Contaminants of Emerging Concern, PFOS should not be missing. This will generate more interest. The paper would also benefit by a conclusion with some comments on the “best” electrochemical methods and maybe comment on any commonality between the contaminants and the detection to guide future research. Finally, some comparison between other methods would be useful. The benefit of electrochemical detection is mentioned but not in contrast to a specific spherics. For example, LC-triple quad verses an electrochemical cell – noting the difference and relative benefit. The paper would benefit from some insight into the commonality of the electrochemical methods for emerging contaminants and contrast to other mythology. This is important for a review paper.

Reply: PFOS has been involved in the revised manuscript as discussed in section 3.7 as below:

PFAs are aliphatic substances containing the moiety CnF2n+1 within the structure, where n≥1. This group also include substances where a perfluorocarbon chain is connected with functional groups on both ends, aromatic substances that have perfluoroalkyl moieties on the side chains, and fluorinated cycloaliphatic substances [136]. PFAs are known as ‘forever chemicals’ as they resist breakdown under natural conditions and has a half-life of >92 years in water. These compounds are used in synthesis of packaging products, carpets, firefighting foams, paints, semiconductors, carpets, etc. The fluorinated regions in PFAs offer unique properties, such as thermal stability due to the presence of strong C-F bonds, stability against acid, bases, oxidizing and reducing agents, oil and water repellence [137]. Toxic effects of PFAs include immunological, cardiovascular, reproductive, developmental and liver effects [138]. Perfluorooctanoic acid (PFOA), perfluorooctane sulfonate (PFOS), GenX, fluorotelomer alcohols (FTOHs), polyfluoroalkyl phosphates are common PFAs. The Heads of the Environmental Protection Agencies for Australia and Newzeland (HEPA) set the PFA guideline values as 0.00023 μg/kg for PFOS and 19 μg/kg for PFOA in aquatic environment [139]. An electrochemical sensor using Hf-doped WO3 was synthesized by a hydrothermal method modified on CPE for detection of PFOA [140]. The higher surface area of the modifier improved the electrocatalytic activity. The limit of detection and linear range were found out to be 1.83 × 10-8 M and 7.0 × 10-8 M to 3.0 × 10-4 M, respectively, using SWV technique. A nanocomposite of WS2-MWCNT was also used for the electrochemical detection of PFOA as well as for supercapacitor application [141]. The synergistic effects of MWCNT and WS2 enhanced the electronic conductivity and sensitivity for detection of PFOA, where the sensor yielded a limit of detection of 2.404 pM.

Also, electrochemical sensors, optical sensors and LC-MS techniques have been compared in Table 2 in the revised manuscript.

Reviewer 2 Report

Comments and Suggestions for Authors

The manuscript provides an overview of publications on new electrochemical sensors that are designed to detect the most common pollutants. Among the pollutants there may be medicines, banned chemicals, pesticides and other products widely used in everyday life. It seems that this work is important for numerous researchers in the field of ecology.

Comments and suggestions:

1. In my opinion, it is necessary to compare the sensitivity to the concentration of pollutants of new electrochemical sensors and traditional techniques. This can be done by introducing an additional column in table 2.

2. In most cases, the mechanisms that cause a sensory response to pollutants are not described.

3. Are the sensors described in the manuscript disposable, or can they be used repeatedly?

Author Response

The manuscript provides an overview of publications on new electrochemical sensors that are designed to detect the most common pollutants. Among the pollutants there may be medicines, banned chemicals, pesticides and other products widely used in everyday life. It seems that this work is important for numerous researchers in the field of ecology.

Comments and suggestions:

  1. In my opinion, it is necessary to compare the sensitivity to the concentration of pollutants of new electrochemical sensors and traditional techniques. This can be done by introducing an additional column in Table 2.

Reply: A new column has been added in the revised manuscript for sensitivity in Table 3 (the then Table 2) to compare the sensitivity to concentration of pollutants.

  1. In most cases, the mechanisms that cause a sensory response to pollutants are not described.

Reply: General mechanism of sensory response to pollutants is described in section 3 as below:

In electrocatalyst based sensors, the electrocatalyst coated on the surface of working electrode undergo redox reaction with the analyte of interest (CECs). The redox reaction between CEC and electrocatalyst results in corresponding electrical signal, from which quantitative and qualitative information about the CEC can be deciphered.

  1. Are the sensors described in the manuscript disposable, or can they be used repeatedly?

Reply: SPEs are disposable, whereas GCE can be used repeatedly after polishing as described in section 3 in the revised manuscript as below:

Commonly used working electrodes are glassy carbon electrode (GCE) and screen printed electrode (SPE). SPE which perform as working electrode alone and one that combines all the three electrodes into a single substrate are available. Features, like portability, low cost, disposability, and omission of electrode polishing, have made SPE popular. SPE also prevents any possible fouling from analytes used previously as they are disposable [144].

Reviewer 3 Report

Comments and Suggestions for Authors

This manuscript was well-written and covered the performance of electrochemical biosensors for CECs in depth. It should be included a table that compares the benefits and drawbacks of these electrochemical biosensors to those of other optical biosensors. Based on this table, the authors should underline why electrochemical biosensor for CECs is superior to others, as well as describe how scientists might enhance these approaches to get the greatest results and application. These methods' perspectives should be thoroughly examined.

Author Response

This manuscript was well-written and covered the performance of electrochemical biosensors for CECs in depth. It should be included a table that compares the benefits and drawbacks of these electrochemical biosensors to those of other optical biosensors. Based on this table, the authors should underline why electrochemical biosensor for CECs is superior to others, as well as describe how scientists might enhance these approaches to get the greatest results and application. These methods' perspectives should be thoroughly examined.

Reply: Comparison of electrochemical sensors, optical sensors, and traditional method like LC-MS is given in Table 2 in the revised manuscript.

Round 2

Reviewer 2 Report

Comments and Suggestions for Authors

In my opinion, in its present form, the manuscript meets the requirements of the journal Molecules